behaviour

imidacloprid, behaviour, locomotor, activity, bee, learning

**Author for correspondence:**
Felicity Muth
e-mail: felicity.muth@austin.utexas.edu

# No evidence for neonicotinoid preferences in the bumblebee *Bombus impatiens*

Felicity Muth[1], Rebekah L. Gaxiola[2] and Anne S. Leonard[3]

[1]Department of Integrative Biology, University of Texas at Austin, TX 78712, USA
[2]School of Biological Sciences, Washington State University, Vancouver, WA 99164, USA
[3]Department of Biology, University of Nevada, Reno, NV 89557, USA

FM, 0000-0003-0904-0589

Neonicotinoid pesticides can have a multitude of negative sublethal effects on bees. Understanding their impact on wild populations requires accurately estimating the dosages bees encounter under natural conditions. This is complicated by the possibility that bees might influence their own exposure: two recent studies found that bumblebees (*Bombus terrestris*) preferentially consumed neonicotinoid-contaminated nectar, even though these chemicals are thought to be tasteless and odourless. Here, we used *Bombus impatiens* to explore two elements of these reported preferences, with the aim of understanding their ecological implication and underlying mechanism. First, we asked whether preferences persisted across a range of realistic nectar sugar concentrations, when measured at a series of time points up until 24 h. Second, we tested whether bees' neonicotinoid preferences were driven by an ability to associate their post-ingestive consequences with floral stimuli such as colour, location or scent. We found no evidence that foragers preferred to consume neonicotinoid-containing solutions, despite finding effects on feeding motivation and locomotor activity in line with previous work. Bees also did not preferentially visit floral stimuli previously paired with a neonicotinoid-containing solution. These results highlight the need for further research into the mechanisms underlying bees' responses to these pesticides, critical for determining how neonicotinoid-driven foraging preferences might operate in the real world for different bee species.

## 1. Introduction

Pollinator declines are thought to be driven in part by the direct and indirect effects of increased exposure to pesticides [1].

Neonicotinoids have been an area of particular concern, including being recently banned in the European Union (EU Press Release, 27 April 2018). As systemic pesticides, they are often found in the nectar and pollen of both crop plants [2] and wildflowers in the vicinity of target applications (via dust from seed treatments or runoff [3]). When consumed, neonicotinoid pesticides can have a number of negative effects on bee colony growth and queen production [4–6], and have also been the focus of substantial study for their sublethal behavioural effects on bees [7,8]. For example, after exposure, bees may be impaired in their motivation to forage [9–11] and collect less pollen [4,12–15] (but see [16]). Bees may also be impaired in their ability to efficiently handle flowers [16] and learn floral associations [17] (but see [10,11,18]). Some of the clearest effects of neonicotinoids on bees' behaviour are disruptions to locomotor activity [19–23], as well as to feeding motivation [23–27].

Most of the research on how neonicotinoids affect bee behaviour has been conducted in laboratory settings [8]. Laboratory-based experiments are often more practically feasible than field-based research, and can allow for specific behavioural effects to be more easily identified. At the same time, it is important that laboratory-based experiments use pesticide dosages that accurately represent bees' exposure, as effects often depend on both the dose and duration. Thus, the question of whether controlled laboratory experiments reflect realistic exposure scenarios is critical for predicting real-world impacts.

One variable likely to impact exposure likelihood is whether bees actively avoid or preferentially visit flowers containing neonicotinoids. This possibility was raised by two recent studies [27,28]. In the first of these studies, Kessler et al. [27] offered individual workers (Bombus terrestris and Apis mellifera) three feeding tubes, each containing water, sucrose or sucrose + a neonicotinoid. At the end of the 24 h trial, more solution was consumed from the tubes containing the neonicotinoid, but the effect depended on neonicotinoid identity and concentration. In a second study, Arce et al. [28] exposed colonies of free-foraging bumblebees (B. terrestris) to six feeders of three possible neonicotinoid concentrations, for 6 h d$^{-1}$ for 10 days. Contrary to Kessler et al., Arce et al. found that at 24 h, there was no preference between control and 11 ppb solutions, and that bees consumed less of 2 ppb solutions. However, across the testing period of 10 days, individuals increased their consumption of solutions of 2 and 11 ppb at a higher rate than control solutions. These, and indeed nearly all experimental studies on bees to date, involve offering bees simple sucrose solutions with a single pesticide of interest added to simulate doses encountered in floral nectar. In reality, floral nectars are chemically complex, for example, also containing secondary compounds such as alkaloids, iridoid glycosides and phenolics [29]. The limited amount of research into how bees respond to variation in more than one nectar trait suggests that preference or aversion can strongly depend on chemical context. For example, bumblebees' responses to nectar containing high levels of an alkaloid depend on sucrose concentration [30], and bumblebees' preference for a pollen fatty acid in solution disappears once sucrose is present [31]. Given the wide range of sugar concentrations of floral nectars (approx. 5–70%, with bumblebees preferring flowers in the 30–55% range [32,33]), we know little about how neonicotinoid preferences identified previously [27,28] might play out across a range of ecologically relevant nectar sugar concentrations.

An additional component of the previously reported preference for neonicotinoids [27,28] that remains unknown is the underlying mechanism. In both Kessler et al. and Arce et al. studies, the mechanism underlying the preference for neonicotinoids is unclear, although electrophysiological results from Kessler et al. [27] indicate that preference driven by oral taste seems unlikely. Likewise, because neonicotinoids are not volatile at room temperature [34], it also seems unlikely that olfactory cues drive preferences. Rather than taste or olfaction, both studies suggest the possibility that preferences may arise through some form of post-ingestive effect. For example, a neonicotinoid might enhance learning of the spatial location of the neonicotinoid-containing 'floral' source through its action on nicotinic acetylcholine receptors (nAChRs) in the bee brain [35–38]. It is also possible that bees prefer neonicotinoid-containing solutions through other post-ingestive means. After all, animals are widely able to associate positive or negative consequences of ingesting a nutrient or toxin with food-related stimuli [39–42]. However, the possibility that preference for a neonicotinoid is driven by its post-ingestive consequences has not been directly explored.

Using concentrations of the neonicotinoid imidacloprid that we confirmed had effects on bumblebees' feeding motivation and locomotor activity (electronic supplementary material, experiments S1 and S2), we asked how bees' relative preferences for imidacloprid in solution depended upon sucrose concentration (experiment 1; figure 1). By closely measuring bees' consumption of solutions across time, we also aimed to gain insight into when preferences for a particular solution emerge, and thus how readily preferences might transfer to a natural foraging scenario. Presumably, the longer any preferences take to emerge, the more difficult it would be for bees visiting multiple floral types or patches to be able to associate the source of the pesticide-laced nectar with its post-ingestive effects [43,44].

| experiment | measure (s) | neonic. dose | sucrose vol. and conc. (% w/w) | protocol/ behaviour measured |
|---|---|---|---|---|
| *1 | preference | 0 versus 0.25, 1 or 10 µg kg$^{-1}$ | 0, 5, 15 and 30% | solution consumed from each feeding tube, measured across time up to 24 h |
| 2 | learned preference | 0 or 0.1 ng (paired with each stimulus) | 10 µl of 30% | bee presented with stimulus + solution, 1 h$^{-1}$ for 6 h, followed by a test (choice between stimuli) |
| *S1 | feeding motivation preference | 0.25, 1, 5 or 10 µg kg$^{-1}$ (for prefs, all versus 0) | 30% | solution consumed from each feeding tube, measured across time up to 24 h |
| S2 | locomotor activity | 0, 0.2, 0.4, 1.1 or 2.2 ng | 2 µl of 30% | no. times bee crosses a centre line over a 2 h period |
| *S3 | preference | 0 versus 0.25 µg kg$^{-1}$ | 15% | solution consumed from each feeding tube at 24 h |
| S4 | learned preference | 0 | 10 µl of 10 versus 50% | proof of concept that bees can learn an association when tested in a protocol similar to Expt. 2 |

*In these experiments the total volume consumed was determined by the bee; in the other experiments bees were fed a specific volume

**Figure 1.** Summary of all experiments.

After not finding evidence for neonicotinoid preferences in this experiment (experiment 1) or in other attempts (electronic supplementary material, experiments S1 and S3), we hypothesized that the lack of preference may have been because bees were unable to associate post-ingestive effects with the location of the solution containing the neonicotinoid. To directly test whether neonicotinoids can have post-ingestive impacts (positive or negative) on the learning of associated stimuli, we carried out an experiment where bees were presented with temporally separated, alternating imidacloprid-containing and control solutions paired with salient stimuli (experiment 2). If neonicotinoids either directly enhance learning of conditioned stimuli or have positive post-ingestive consequences for bees, we expected that they would preferentially visit stimuli paired during training with a neonicotinoid-containing solution. Conversely, if consuming neonicotinoids has detrimental post-ingestive consequences (in the same way as toxins in food [39] or plant-derived defensive compounds [45]), we expected that bees would avoid the stimuli previously paired with the neonicotinoid-containing solution.

## 2. Methods

### 2.1. Subjects, colony maintenance and general methods

Bumblebee colonies (*Bombus impatiens*, 50–70 individuals colony$^{-1}$ with natal queen) were purchased from Koppert Biological Systems (MI, USA). They were maintained on approximately 0.5 g of honeybee-collected pollen (Koppert Biological Systems), placed directly into colonies every 1–2 days. Depending on the experiment, a given colony was connected to one of three foraging arenas (ranging in size from 13 to 16 m$^3$), lit from above by an LED light strip (2100 lumens, 4000 K, Lithonia Lighting, Conyers, GA, USA); the room was illuminated by both fluorescent and natural light. Bees could free-forage on 15% (w/w) sucrose solution provided ad libitum via a white cotton-wicked feeder in each arena. We chose this concentration of sucrose to be low enough that bees would still be motivated to consume sucrose in the experimental assay, but high enough that colonies would remain productive when maintained on it. We collected bees from these sucrose feeders upon landing using an insect aspirator (Bioquip Products, CA, USA) for use in experiments. We then cold-anaesthetized bees, and transferred them to a preference tube (experiment 1) or container (experiment 2); details in sections below. For all experiments, we used bees from multiple colonies, represented equally across treatments within each experiment (experiment 1 $n = 5$ colonies; experiment 2 $n = 2$ colonies), and inter-colony variation was always accounted for in analyses.

**Table 1.** Final sample sizes for experiment 1 by treatment (12 total) (summary of excluded bees by treatment shown in electronic supplementary material, table S2). (We tested bees' preferences for imidacloprid (IMD) across a range of sucrose and pesticide concentrations, across a series of time points. Bees were presented with two feeding tubes: one containing sucrose with imidacloprid present and the other containing sucrose of the same concentration but with no imidacloprid present. To see values in ppb, see the electronic supplementary material, table S3.)

| IMD concentration (μg kg⁻¹) | sucrose concentration (%) | | | |
| --- | --- | --- | --- | --- |
| | 0 | 5 | 15 | 30 |
| 0.25 | 13 | 31 | 33 | 34 |
| 1 | 20 | 26 | 35 | 36 |
| 10 | 16 | 28 | 32 | 34 |

## 2.2. Pesticide solutions

To make up pesticide solutions for use in experiments, 93.00 mg of analytical standard PEDESTAL® imidacloprid powder was dissolved in 93 ml of acetone. Aliquots of this solution were then added to sucrose solution (concentration dependent on the experimental treatment) for use in experiments. To make control solutions, we added the same amount of acetone to the same volume and concentration of sucrose. Solutions were stored in amber bottles in a refrigerator at 4°C, and were always presented to bees immediately after being poured from these bottles (solutions then immediately returned to the refrigerator). Fresh solutions were made every 4–5 days.

## 2.3. General protocol for data analyses

All analyses were carried out in R v. 3.5.1. [46]. To carry out generalized linear mixed models (GLMMs), we used the glmer() function in the lme4 package [47] and to carry out linear mixed models (LMMs), we used the lme() function in the nlme package [48]. In cases where our models did not generate *p*-values (i.e. for GLMMs), we compared the fit of models using the anova() function to carry out a likelihood ratio (LR) test between models with and without the variable in question. In all cases, we ran full models initially, before removing non-significant interaction terms. To determine the direction of interaction effects and to carry out *post hoc* tests on GLMMs to determine where significance lay between treatments, we used the packages effects() [49] and emmeans() [50]. Details on analyses for each experiment are given in the subsections below.

## 2.4. Experiment 1: imidacloprid preferences across a range of sucrose and pesticide concentrations

### 2.4.1. Methods

After establishing effects on feeding motivation (electronic supplementary material, experiment S1 and figure S4) and locomotor activity (electronic supplementary material, experiment S2 and figure S5) in line with previous studies, we sought to explore preferences for neonicotinoids across a range of imidacloprid and sucrose concentrations (12 treatments total; treatments and final sample sizes shown in table 1). To do this, we placed individual bees in transparent plastic cylindrical preference tubes with ventilation holes (TAP plastics, USA; length × diameter 13 × 2.5 cm, wall thickness: 1.6 mm), sealed at both ends with rubber stoppers. After a 2 h acclimatization period, we removed one of the rubber stoppers, and replaced it with two glass feeding tubes (internal diameter × length: 3.4 × 150 mm, World Precision Instruments, USA) fitted into a plug (figure 1). The cotton-plugged feeding tubes were spaced 5 mm apart and each filled with 1000 μl of a given solution. We measured how much of each solution bees consumed (the distance the meniscus migrated in each tube from its marked initial point) at the following time intervals: 15, 30, 60, 90 min and 3, 4, 5, 7 and 24 h (1 mm = 9.079 μl). We chose these times based on previous work showing that they were informative for comparing consumption over time [31]. In this experiment, the two tubes always contained different solutions: one held the imidacloprid-containing solution and the other held the control solution for that treatment (i.e. the same concentration of sucrose, but without

imidacloprid present). The amount of acetone used in control solutions always matched the amount used in imidacloprid-containing treatments (12 different control solutions in total). The imidacloprid concentrations we used were both within the range of what has been used previously [27] and of what has been found in the nectar of flowers in the field [2,51] (see also the electronic supplementary material, table S2 in [17]). We chose sucrose concentrations within the range of natural flowers that bumblebees visit [32]. We also included water because bees will consume water, and because previous work indicates that preferences may exist for reward chemicals in water that disappear once sucrose is present [31]. Previous work addressed preferences at 0.5 M (approx. 17%) [27] and at 30% sucrose [28].

Alongside the experimental treatments, we also carried out controls to account for evaporation which could differ depending on sucrose and pesticide concentration: we set up preference tubes in the same way as described above, but without a bee. We collected data for 8–13 replicates for each of the 24 solutions (sample sizes in the electronic supplementary material, table S1). After finding that solutions did not differ in evaporation over time depending on imidacloprid concentration, we pooled the evaporative controls for each sucrose concentration, and subtracted the mean evaporation at each time point from the raw data (electronic supplementary material, figure S1; details in the electronic supplementary material).

We tested 30–38 bees in each treatment initially, but excluded all bees that had not consumed a solution at 7 h, or consumed less than 2 mm after controlling for evaporation (18.19 µl) at 24 h; we chose this cut-off because bees that had discovered a solution would typically drink more than this, so it was deemed that such a small amount 'consumed' was possibly owing to error (leaking). The resulting sample sizes are shown in table 1. Bees that were excluded were mostly from the water and 5% sucrose treatment groups (summary of excluded bees by treatment is shown in the electronic supplementary material, table S2).

### 2.4.2. Data analyses

To determine whether bees preferred neonicotinoid-containing solutions over control solutions, and whether this differed by imidacloprid concentration, sucrose concentration, and over time, we carried out an LMM (model 1) with the response variable 'amount consumed' (of each specific solution), using the non-cumulative amount consumed since the previous time point measured, and the explanatory variables: 'sucrose concentration' (ordinal factor, four levels), 'imidacloprid concentration' (ordinal factor, three levels), 'solution type' (neonicotinoid or control), 'time' (continuous variable) and the random factor 'bee' nested in the random factor 'colony'. We limited this analysis to the data from five time points: 90, 180, 240, 300 and 420 min. We did this because at time points previous to this, many bees had not consumed anything, resulting in residuals from models which were greatly skewed, and which could not be transformed to be used in a parametric analysis. Similarly, the consumption data at 24 h were much larger than the rest of these values, and so we analysed these data separately (total amount consumed at 24 h), carrying out a single LMM (model 2) with the same explanatory variables described in model 1, except for the variable 'time'. We also tested for differences in the cumulative amount consumed (of each solution) at 90 min using an LMM (model 3), again, using the same explanatory variables as model 1 except for 'time', to determine whether there were differences early on that then disappeared across the course of testing.

### 2.4.3. Results

When comparing consumption of solutions across a series of specific time points (from 90 min to 7 h), we found no overall preference for imidacloprid versus control solutions (model 1: $F_{1,3029} = 0.62$; $p = 0.43$; figure 2). Instead, the total amount consumed was explained by the three-way interaction between time, imidacloprid concentration and sucrose concentration ($F_{6,3029} = 2.15$; $p < 0.05$): the amount bees consumed decreased over time ($F_{1,3029} = 190.78$; $p < 0.0001$; electronic supplementary material, figure S2), but this effect depended on the concentration of sucrose and imidacloprid. Specifically, bees consumed least in the water treatment, and most in the 15% sucrose treatment ($F_{3,322} = 26.31$; $p < 0.0001$; electronic supplementary material, figure S2), and this interaction further depended on imidacloprid concentration, with the higher-dose treatment of 10 µg kg$^{-1}$ suppressing feeding most in the 15% sucrose treatment (unsurprising since bees would have consumed the most neonicotinoid in this treatment).

Because the amount of solution consumed varied greatly across sucrose concentration treatments (potentially masking some of the variation within each of these treatments), we additionally ran separate models within each sucrose concentration treatment (including the same variables as model 1 except for 'sucrose concentration'), in order to clarify whether preferences might exist within each of these groups (results in table 2).

(*a*) water versus water + IMD

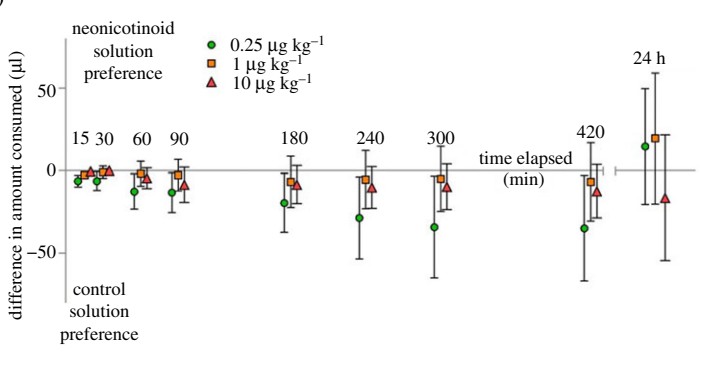

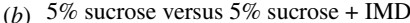

(*b*) 5% sucrose versus 5% sucrose + IMD

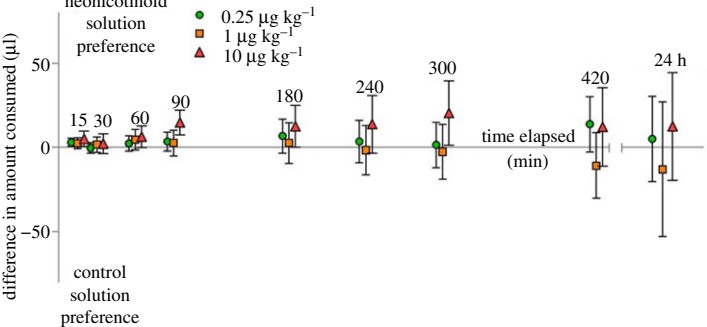

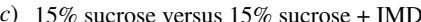

(*c*) 15% sucrose versus 15% sucrose + IMD

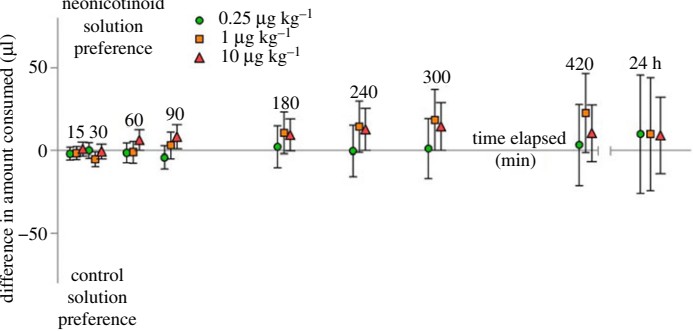

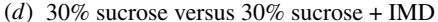

(*d*) 30% sucrose versus 30% sucrose + IMD

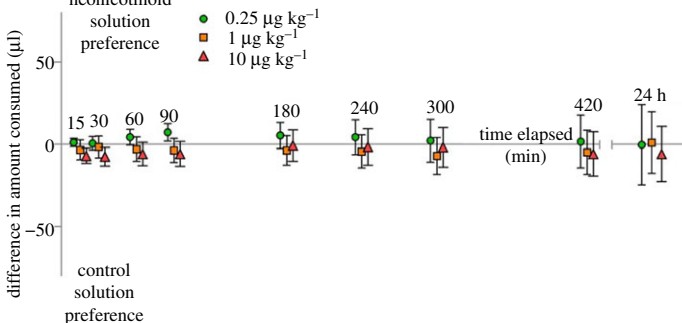

**Figure 2.** The relative preference for the neonicotinoid-containing solution relative to control solution (shown as the difference in the amount consumed of these two solutions) over time for (*a*) water, (*b*) 5% sucrose, (*c*) 15% sucrose and (*d*) 30% sucrose. Within each graph, data are shown for the three different imidacloprid (IMD) concentrations; these are offset on the *x*-axis to make data easier to visualize but were all measured at the same time points. Data shown are means ± s.e.m.

When we separately addressed whether preferences might have emerged by 24 h (the time point at which preferences were measured in [27]), we found no evidence for a preference for a neonicotinoid-containing solution over control solutions (model 2: $F_{1,337} = 0.54$; $p = 0.46$). There was a significant interaction between the concentration of sucrose and the concentration of neonicotinoid ($F_{6,322} = 2.45$;

**Table 2.** Results of the four models (one for each sucrose concentration treatment) addressing which factors affected the consumption of the solutions over time; significant effects are shown in bold.

| question addressed | water | 5% | 15% | 30% |
|---|---|---|---|---|
| did bees consume more of the control or neonicotinoid-containing solutions? | **preference for control solution** $(F_{1,439} = 4.05; p < 0.05;$ figure 2*a*; **electronic supplementary material, figure S2a-c)** | no preference $(F_{1,763} = 1.54;$ $p = 0.22;$ figure 2*b*; electronic supplementary material, figure S2d–f) | no preference $(F_{1,896} = 2.59; p = 0.11;$ figure 2*c*; electronic supplementary material, figure S2 g–i) | no preference $(F_{1,934} = 0.20;$ $p = 0.66;$ figure 2*d*; electronic supplementary material, figure S2j–l) |
| was there an effect of time on the amount consumed? | **bees consumed less over time** $(F_{1,439} = 9.54; p < 0.005)$ | **bees consumed less over time** $(F_{1,763} = 50.29; p < 0.0001)$ | **bees consumed less over time** $(F_{1,896} = 76.65; p < 0.0001)$ | **bees consumed less over time** $(F_{1,934} = 132.03;$ $p < 0.0001)$ |
| was there an effect of imidacloprid concentration on the amount consumed? | no effect $(F_{2,42} = 0.10; p = 0.90)$ | no effect $(F_{2,78} = 0.12; p = 0.88)$ | **imidacloprid suppressed feeding the most in the highest-dose treatment (imidacloprid conc. × time:** $F_{2,896} = 7.33; p < 0.001).$ | no effect $(F_{2,97} = 1.00; p = 0.37)$ |

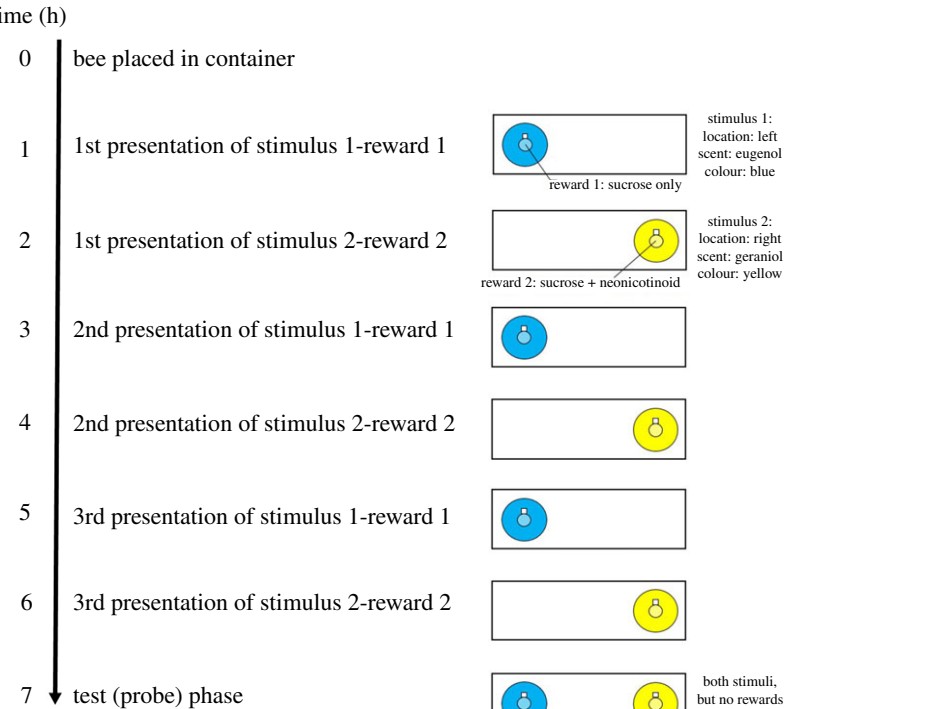

time (h)

| | |
|---|---|
| 0 | bee placed in container |
| 1 | 1st presentation of stimulus 1-reward 1 |
| 2 | 1st presentation of stimulus 2-reward 2 |
| 3 | 2nd presentation of stimulus 1-reward 1 |
| 4 | 2nd presentation of stimulus 2-reward 2 |
| 5 | 3rd presentation of stimulus 1-reward 1 |
| 6 | 3rd presentation of stimulus 2-reward 2 |
| 7 | test (probe) phase |

stimulus 1:
location: left
scent: eugenol
colour: blue

reward 1: sucrose only

stimulus 2:
location: right
scent: geraniol
colour: yellow

reward 2: sucrose + neonicotinoid

both stimuli,
but no rewards
(water only)

**Figure 3.** Experimental design of experiment 2. The specific location, scent and colour of stimuli shown here are a single example; in the experiment, eight possible combinations were used. Similarly, the order that the reward type was presented (sucrose or sucrose + neonicotinoid first) was represented equally across treatments.

$p < 0.05$): bees generally consumed least in the water treatment, and most in the 15% sucrose treatment, while at the highest dose of 10 µg kg$^{-1}$, bees consumed a similar quantity across all sucrose concentration treatments (electronic supplementary material, figure S3).

Finally, to address the possibility that preferences at 90 min (the first time point at which bees had consumed enough to be statistically assessed using a parametric model) might be masked in our analyses if they subsequently disappeared over time, we separately analysed the data at 90 min. Again, we found that bees did not consume more of the neonicotinoid-containing solution relative to the control solution (model 3: $F_{1,337} = 0.35$; $p = 0.56$) and that they consumed least in the water treatment, and most in the 15% sucrose treatment ($F_{3,328} = 21.32$; $p < 0.0001$). Bees did not differ in how much they had consumed overall at 90 min depending on imidacloprid concentration ($F_{2,328} = 1.32$; $p = 0.27$).

## 2.5. Experiment 2: do bees develop preferences for or against a neonicotinoid based on its post-ingestive effects?

### 2.5.1. Methods

We aimed to determine whether bees would preferentially approach or avoid a stimulus paired with a neonicotinoid solution when presentations were temporally separated, allowing for post-ingestive effects to be associated with the stimulus. To do this, we carried out an experiment where bees were presented with neonicotinoid-containing and control solutions, each paired with a particular stimulus, with a delay between presentations (diagram of experimental design in figure 3). In this experiment, bees ($n = 76$) were individually held in plastic preference containers (Rubbermaid TakeAlongs®, USA; length × width × height = 22 × 15 × 5.5 cm, lid centre replaced with ventilation mesh). After 1 h, we presented each bee with a stimulus (consisting of three components) paired with either 10 µl of 30% sucrose with no imidacloprid, or with imidacloprid added (a dose of 0.1 ng). We chose 30% sucrose because it is a concentration that we knew would readily be consumed by bees. To mimic a scenario in which bees encounter two readily distinguishable flower types differing in neonicotinoid presence, we used floral stimuli that a supplementary experiment (electronic supplementary material, experiment S4) demonstrated that bees would readily learn to distinguish. These floral stimuli differed in three components [52]: colour (blue or yellow), scent (eugenol or geraniol) and location (left or right). We always presented the same

combination of stimuli for each bee, but we used eight possible combinations of these three stimuli across bees. We chose stimuli that varied in multiple aspects to make it as easy as possible for bees to learn associations between the stimuli and outcome. While this meant that we could not address possible interactive effects of stimuli on preference, we included the same combinations of stimuli across both treatments to control for preferences being driven by particular stimuli. These stimuli were spaced 45 mm apart, and constructed from a laminated strip of paper printed with a coloured disc (20 mm diameter). A small square of filter paper (approx. 5 × 5 mm) was taped next to each coloured disc, onto which we pipetted 1 µl of scent (diluted 1 : 300 in mineral oil). We pipetted 10 µl of the treatment or control solution directly onto the laminated coloured disc, and observed the bee consume it. Directly after consumption, we removed the stimulus from the preference container. If bees did not move upon a stimulus presentation, we stimulated their antennae with 30% sucrose on a toothpick to induce them to feed. All bees included in the final dataset consumed the solution presented to them on all trials.

We presented each bee with a stimulus–reward pair every hour for 6 h, alternating between neonicotinoid-containing and control solutions (i.e. three presentations per stimulus–reward type pair; figure 3). We used this inter-trial interval in order to give the bee time to associate the post-ingestive effects of a solution with the cues paired with it, as this time period ensures maximal pesticide absorption [53–55]. We alternated which solution was given to the bee first, with $n = 30$ bees receiving the neonicotinoid-containing solution first and $n = 46$ receiving control solution first.

One hour after the final presentation, we gave bees a test (probe) trial (electronic supplementary material, figure S7). In the test phase, bees were presented with both stimuli they had previously encountered, spaced 45 mm apart, each containing 10 µl water. During the test trial (but not stimulus presentation trials), we filmed the bee from above for 5 min. From this video, we coded the total time they spent on each stimulus in the five-minute observation period (using the program Solomon Coder, solomoncoder.com). Videos were coded by an observer blind to treatment. We then used data from the first 5 min of the observation to determine if bees had a preference; a supplementary experiment (electronic supplementary material, experiment S4) confirmed that preferences could be detected just as strongly both after 1 min and after 5 min of observation. We did not address the first colour bees probed because electronic supplementary material, experiment S4 showed that this was not a useful measure of preference. Because this was a new protocol, electronic supplementary material, experiment S4 also served to confirm that bees could learn to discriminate between the conditioned stimuli on the basis of differences in sucrose concentration (10 versus 50%) under the same training conditions (electronic supplementary material, experiment S4). While the rewards used in electronic supplementary material, experiment S4 could be discriminated based on taste rather than post-ingestive consequences, the strong effects that we found using this protocol, combined with other evidence that bees are able to learn based on post-ingestive feedback on shorter timescales [31] supported the idea that this protocol would capture learning based on post-ingestive effects of neonicotinoids (which are absorbed within 1 h [53–55]).

### 2.5.2. Data analyses

To determine whether bees preferred the stimulus that had previously been paired with either the neonicotinoid-containing solution or the control solution, we carried out an LMM (model 4) with the response variable being the total time on the stimulus (seconds) and the following explanatory factors: 'solution type' (neonic or control); 'colour paired with neonic' (blue or yellow); 'scent paired with neonic' (geraniol or eugenol); 'location of neonic' (left or right); order of presentation (neonic or control solution first) and the random factor 'bee' nested in the random factor 'colony'.

### 2.5.3. Results

Bees did not prefer a particular stimulus based on the solution type it had previously been paired with: during the test phase, bees did not spend more time on either the stimulus that had previously been paired with the neonicotinoid-containing solution or the stimulus paired with the control solution (model 4: $F_{1,34} = 0.05$; $p = 0.94$; figure 4).

## 3. Discussion

Despite being recently banned in the EU, neonicotinoid pesticides are still widely used in many parts of the world, including the USA and China. They have been of intense concern in terms of their effects on bees, however some of the more basic aspects of their effects on behaviour still remain unclear, especially

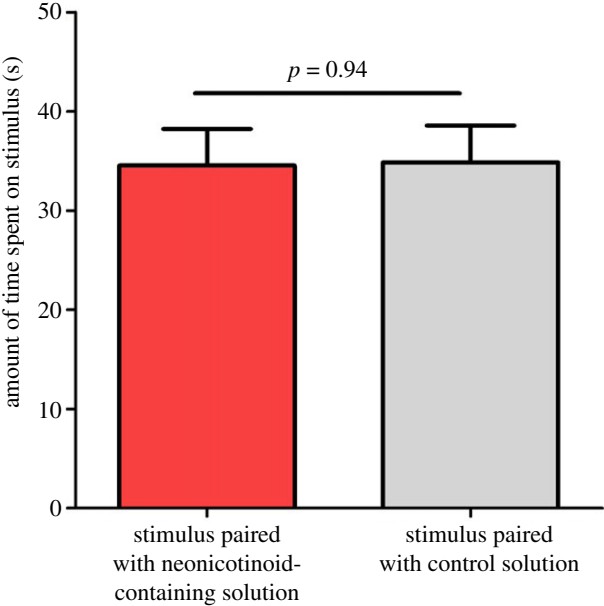

**Figure 4.** The amount of time (mean ± s.e.m.) during the 5 min test phase that bees spent on either the stimulus that had previously been paired with a neonicotinoid-containing solution or the stimulus that had previously been paired with a control (sucrose only) solution.

in non-honeybee species [56]. In line with previous studies, we found that the consumption of imidacloprid by bumblebees disrupted feeding motivation and locomotion activity shortly after initial exposure. Contrary to previous findings, we did not find that bees preferred to consume solutions containing neonicotinoid pesticides, at any of the pesticide or sucrose concentrations we tested (or in any of a variety of preference chamber arrangements: figure 1).

## 3.1. No evidence for preferences for neonicotinoid-containing solutions

Without knowing the behavioural mechanism behind the previously found preferences in Kessler *et al.* [27], it is difficult to know why we did not find them here. However, we can suggest a few possibilities. First, the main difference between our study and the previous ones is the species of bumblebee used (*B. impatiens* in the current study compared to *B. terrestris*). Different *Bombus* species could have a number of physiological differences that might give rise to different dose–response curves, including their body size, the sensitivity of their acetylcholine receptors [35,37] or their speed at metabolizing neonicotinoids [23]. Comparing the inter-tegular span (as a proxy for body size [57]) of the bees (*B. impatiens*) we used in experiment 1 with data available on the inter-tegular span in a study that used commercial *B. terrestris* (Dryad Digital Repository dataset from [13]) indicates that commercial *B. terrestris* workers may be larger than commercial *B. impatiens* workers (mean ± s.d. (in mm): *B. terrestris*: 4.10 ± 0.57 (*n* = 340); *B. impatiens*: mean ± s.d.: 3.32 ± 0.25 (*n* = 253)). Given that in Kessler *et al.* [27], the imidacloprid preference was only found at one of the four dose levels tested, it could be that none of the doses we tested struck upon the one that would generate a preference in *B. impatiens*. However, it is worth noting that we did find similar dose-dependent effects to Kessler *et al.* [27] in terms of imidacloprid's effect on feeding motivation, indicating that *B. impatiens* reacts similarly to *B. terrestris* in response to this neonicotinoid in at least one aspect of behaviour.

We currently know little about species- or colony-level differences in sensitivity to pesticides, although studies comparing across genera (e.g. *A. mellifera* versus *B. terrestris*) have found that these bees differ in their sensitivity to and speed at which they metabolize neonicotinoids [23,25]. To our knowledge, the sensitivity of different bumblebee species to neonicotinoids have not been directly compared, and indeed, the vast majority of work that has been done uses only two commercial species, *B. terrestris* and *B. impatiens* [58]. Beyond the clear need to expand genera-level comparisons, a broader species comparison within *Bombus* would be useful, because bumblebees are key for both agricultural and ecosystem services, and appear to be one of the groups of bees more sensitive to neonicotinoid pesticides [59,60].

It is also possible that differences in protocol account for the discrepancy between our and previous results. For example, although it is not clear from Kessler *et al.* [27] how spatially separated feeding tubes

were, bees might be more able to associate post-ingestive effects of a solution with feeding tubes that are more spatially separated than in our experiment. For example, if a neonicotinoid enhances learning of associated cues such as spatial location (as suggested by the authors in [27,28]), this might happen more readily if the two solutions are more clearly spatially identifiable. However, we did not find such an effect either in the electronic supplementary material, experiment S3 where we had a greater distance between feeding tubes (electronic supplementary material, figure S6), or in experiment 2 where we made the stimuli associated with the two solutions salient and distinct (containing colour, scent and location cues), and thus readily learnable (electronic supplementary material, experiment S4). Finally, it is of course possible that differences exist in wild-foraging bees that we did not detect here. While individuals do show clear preferences for solutions using our assay when solutions are readily distinguishable (confirmed with solutions of different sucrose concentration, A. S. Leonard 2013, unpublished data), there may be differences between how bees sample and consume sucrose when contained individually versus free-foraging on flowers [61].

Interestingly, we did find that bees preferred to consume water over water containing imidacloprid. It is possible that bees preferred the taste of the water over the neonicotinoid-containing solution, which was then masked once sucrose was present (as is also the case for fatty acid preferences [31]). Although sugar would always be present in nectar, bees have been found consuming water from rainwater puddles containing neonicotinoids and other pesticides [62]. Future work might determine whether the preference we found here for pesticide-free water plays out in a more natural setting. If so, it might be advantageous to provide commercial bees with pesticide-free water sources.

## 3.2. No evidence for post-ingestive preferences

In the absence of evidence that bees taste or smell neonicotinoid solutions, post-ingestive feedback has been suggested to be a mechanism that could drive bees' preference for neonicotinoid-containing sucrose solutions. However, this possibility has, until now, to our knowledge not been tested. At least in other scenarios, bees are able to associate the consequences of ingesting a solution with paired stimuli. For example, Ayestaran et al. [45] tested honeybees in a scenario in which a scent was first paired with a sugar reward, and then the sugar reward was devalued through being paired with a compound that induced post-ingestional 'malaise'. After this, bees responded less to the original conditioned scent compared to controls [45]. In experiment 2, we failed to find evidence that bees, at least under the conditions of this experiment, formed associations between neonicotinoid-containing solutions and a combination of salient stimuli, based on the post-ingestive effects of the neonicotinoid. We also did not find evidence that neonicotinoid consumption enhanced learning of cues paired with it (as suggested by the authors in [27,28]), and as has been shown to be the case for nicotine in floral nectar under experimental conditions [63].

Why did bees not form post-ingestive preferences? One possibility may be that the post-ingestive effects of imidacloprid at the dose we used were not strong enough to give rise to an association with the cue combinations we presented. We selected this dose (0.1 ng per exposure) because it represents a field-realistic acute exposure that we expected would have behavioural effects based on previous B. impatiens research [10]; in this study, we found behavioural effects at 0.2 ng, but we reduced the dose in the current study because bees were exposed to it three times. Alternatively, bees may not have been able to associate the post-ingestive effects of consuming a dose of imidacloprid with a particular stimulus because the effects they experienced lasted for longer than the time period between association–reward pairings (1 h), meaning that bees were unable to pinpoint which stimulus was causing the post-ingestive effect [43]. Finally, it is possible that imidacloprid impaired learning such that bees were unable to pair the post-ingestive effects with the associated stimulus. While olfactory and spatial learning have been found to be impaired by neonicotinoids [17,53], visual learning generally is not [10,11,18,64], so this explanation seems unlikely, given that bees could have learned to discriminate between the two stimuli exclusively on the basis of colour.

One of the most robust effects found both within our experiment and across related experiments [23–27] relates to the negative effects of neonicotinoid consumption on feeding motivation. This raises the issue of whether it is problematic to determine preferences for a chemical that inhibits feeding motivation by measuring how much an animal consumes of food containing that chemical. For example, in experiments where the bee controls which solutions it samples and the order in which it samples them, we might expect that bees which consumed neonicotinoid-containing solutions first would be less motivated to locate and/or feed from the control solution, compared to bees discovering the solutions in the opposite order, generating an apparent preference. Such effects could be reduced by having the experimenter control solution sampling rather than the bee.

## 3.3. Neonicotinoid effects on feeding motivation and locomotor activity

Despite not finding any preferences or evidence of learning based on post-ingestive effects, we did find clear behavioural effects of imidacloprid on two other aspects of bee behaviour relevant to performance. Bees that consumed higher doses of neonicotinoid were less motivated to feed (electronic supplementary material, experiment S1), in line with previous findings [23–27]. We also found short-term effects on locomotor activity (electronic supplementary material, experiment S2), with the general trend of dosed bees being less active in the short term (less than 60 min after exposure), and more active after a longer period had elapsed (1–2 h). Previous work in honeybees has found that effects on locomotor activity are both dose- and time-dependent: bees given 1.25 ng of neonicotinoid were more active at the three time points sampled (15, 30 and 60 min), whereas bees given higher doses (2.5–20 ng) were generally less active after 30 and 60 min [20]. With regard to bumblebees, we know of no studies prior to this one that have addressed acute effects of a neonicotinoid to locomotor activity. A study of the long-term effects on activity, by contrast, found no effect of imidacloprid on bumblebee activity 4 days after being dosed [25], while another found that imidacloprid decreased the daily activity of bees across a number of days [23]. Disruption to locomotor activity through either hypoactivity or hyperactivity may explain some of the detrimental effects found on foraging and flower-handling ability in free-flying bumblebees [10,13,16].

In conclusion, our results highlight the need for more research into the behavioural mechanism behind previous findings of bees' preference for neonicotinoid pesticides. If the lack of replicability is owing to species differences, or a small difference in protocol, both of these have implications for how we interpret such results in an ecologically realistic setting. Further, establishing clear and replicable behavioural protocols is critical for informing future research into how bees respond to other systemic pesticides introduced as replacements for neonicotinoids [65].

Data accessibility. Data are available from the Dryad Digital Repository: https://doi.org/10.5061/dryad.0gb5mkkxw [66].
Authors' contributions. All authors contributed to experimental design and the writing of the manuscript. R.L.G. collected data for experiments 1 and 2; and electronic supplementary material, experiments S2 and S3. F.M. collected the data for experiment 2; and electronic supplementary material, experiments S1 and S4. F.M. carried out the statistical analyses and drafted the manuscript. All authors gave final approval for publication and agree to be held accountable for the work performed therein.
Competing interests. The authors have no competing interests.
Funding. Funding was provided by the US Department of Agriculture NIFA postdoctoral fellowship and a L'Oréal For Women in Science postdoctoral fellowship (F.M. and R.L.G.). A.S.L. was funded by the USDA (AWD no. 2018-67014-27543).
Acknowledgements. We thank the Leonard Laboratory (J. Francis, D. Picklum, S. Richman and A. Tatarko) and Maj Rundlöf for helpful comments on earlier versions of the manuscript, and Matt Forister and Ali Urza for advice on statistical analysis of data. We also thank Anna Tatarko for assistance collecting data in experiment 2.

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
