## [Reviewer comments · Royal Society Open Science]

Review History

RSOS-191883.R0 (Original submission)

Review form: Reviewer 1

Is the manuscript scientifically sound in its present form?

Yes

Are the interpretations and conclusions justified by the results?

No

Is the language acceptable?

Yes

Do you have any ethical concerns with this paper?

No

Have you any concerns about statistical analyses in this paper?

Yes

Recommendation?

Major revision is needed (please make suggestions in comments)

Comments to the Author(s)

I would first like to disclose that I was not an original reviewer of the previous MS submission. In an attempt to be as unbiased as possible, I assessed the quality of the new version of the MS before looking at the previous reviewer's comments and responses.

My overall view is that this work should definitely be published and I think Roy Soc Open Sci is a good fit for this MS. The amount of work gone into the study is impressive and the presentation of the pilot studies to support the main experiments is great to see.

I do, however, think that whilst the overall results presented are of interest, there are a number of potential caveats that need to be better considered/explained in the MS so that a lay reader can understand the possibilities of why some studies may have found preferences whilst others (such as yours) may not.

Major Comments

Experiment 1

I think the key methodological differences between the Kessler and Arce papers needs to be made clearer. This is important as your 1st experiment is more akin to the Kessler paper in which solitary individuals are used and a preference after 24 hours is studied. Arce actually found avoidance during the first 24 hours and it was only after a number of days and repeated visits (i.e. in the tens of visits per bee) that colony level preferences were starting to manifest. In my view there are issues with the design of placing a worker in solitary confinement for 24 hours to choose between two solutions to feed upon (as Kessler and your study conducts). Firstly, solitary workers do not act normally and have little motivation to forage in a normal manner. Secondly, it is not clear to me how much a worker will repeatedly feed during 24 hours, and from what I can tell (sorry if I have misunderstood) you do not provide any behavioural data to show frequency of feeds from the tubes. Therefore, what does your consumption data actually tell you about how many times a worker needs to sample a tube to reinforce a preference or aversion? This data is needed to elucidate whether a worker in fact finds a tube, feeds on it and is then satiated for a long period of the experiment. If feed frequency is low then are you actually testing a preference appropriately? For instance, they may not have had the chance to sample each tube enough times for any preference to be reinforced, and therefore you may expect a 50:50 outcome, which seems to be what you found. I think you therefore need to analyse some of your data to convince the reader this is not the case. Glancing through your raw data you could perhaps use consumption rate differentials per time point to give you a value on this or perhaps look at last feed or at least highlight this potential caveat in the discussion. If you do not agree with me, then I think your design needs to be better justified somewhere in the methods. Third, following on from this, does having a higher % sugar concentration increase the risk of what I have highlighted above i.e. on drinking a high calorie solution (e.g. 30%) may mean that the worker is less likely to drink again than if presented with low calorie (e.g. 5%) because of the energetic benefit. Was this found? Fourth, if your bees are different sizes then this may also determine the frequency of feeds and hence bring variation into your data. When considering points 2-4 are you able to control for these possible confounding effects?

Experiment 2

Arce findings showed that whilst the preference was significant, the majority of visits to neonic feeders was not comparatively huge but perhaps more subtle. My concern with your 2nd experiment is that there are at least two stimuli that are dramatically different between your feeders associated with treated or untreated solutions. Is it therefore not possible that these conflicting cues could override any avoidance/preference? Also, is six hours enough time to develop this preference and can you add frequency of visits and time per hour for each of the 1st, 2nd and 3rd presentations? You use your Exp S4 to justify, but the difference between 10 and 50% is quite large and a better justification for being able to associate cues with a post-ingest effect using this setup may need to be more clearly explained.

Minor

Line 34 – insert over a 24-hour period.

Line 51 – ‘after exposure bees forage less frequently’ – I am not sure precisely what you mean by this statement.

Line 264-265 – does this suggest you have stressed the bee?

Lines 347-350 – would this not be better placed in the intro?

Line 353 – you use the term doses, but I think it is more accurate you use concentration because you did not consider this as a covariate in your analyses. i.e. consumption per unit gram of mass would better give dosage.

Line 358 – be clear that Arce had a very different experimental setup.

Review form: Reviewer 2

Is the manuscript scientifically sound in its present form?

Yes

Are the interpretations and conclusions justified by the results?

Yes

Is the language acceptable?

Yes

Do you have any ethical concerns with this paper?

No

Have you any concerns about statistical analyses in this paper?

No

Recommendation?

Accept with minor revision (please list in comments)

Comments to the Author(s)

I feel the authors have done a good job of addressing the initial reviewers' concerns with the statistics. It all made sense to me. I see no problems. My assessment is that this paper makes a very important contribution. They have not been able to find a preference for neonicotinoids in *B. impatiens*. That places an informative caveat around the generality of the reports for *B. terrestris*. I feel the discussion in this paper is careful and balanced and demonstrates thorough and sound scholarship. I hope this work will inspire further work into this important issue.

My only concern is very minor. It would seem in revision that some of the reference formatting has gone a bit haywire. The reference format is as a numbered list, but at various points we have references in the format Arce et al (2018). I picked these up at lines 82, 85, 358 and 408, but there may be others. I recommend a careful check of the references before final submission Overall I found this a solid and constructive study.

Decision letter (RSOS-191883.R0)

14-Jan-2020

Dear Professor Muth,

The editors assigned to your paper ("No evidence for neonicotinoid preferences in the bumblebee *Bombus impatiens*") have now received comments from reviewers. We would like you to revise your paper in accordance with the referee and Associate Editor suggestions which can be found below (not including confidential reports to the Editor). Please note this decision does not guarantee eventual acceptance.

Please submit a copy of your revised paper before 06-Feb-2020. Please note that the revision deadline will expire at 00.00am on this date. If we do not hear from you within this time then it will be assumed that the paper has been withdrawn. In exceptional circumstances, extensions may be possible if agreed with the Editorial Office in advance. We do not allow multiple rounds of revision so we urge you to make every effort to fully address all of the comments at this stage. If deemed necessary by the Editors, your manuscript will be sent back to one or more of the original reviewers for assessment. If the original reviewers are not available, we may invite new reviewers.

- Data accessibility

<http://datadryad.org/submit?journalID=RSOS&manu=RSOS-191883>

- Competing interests

- Authors' contributions

- Acknowledgements

- Funding statement

on behalf of Dr Jake Socha (Associate Editor) and Kevin Padian (Subject Editor)
openscience@royalsociety.org

Associate Editor's comments (Dr Jake Socha):

Both reviewers were enthusiastic about the importance of this study and found it to be a strong piece of scholarship. However, reviewer 1 provided some substantial comments that need to be considered. Reviewer 2's comments were relatively minor and should be easily addressed. We look forward to receiving your revision.

Comments to Author:

Reviewers' Comments to Author:

Reviewer: 1

Comments to the Author(s)

I would first like to disclose that I was not an original reviewer of the previous MS submission. In an attempt to be as unbiased as possible, I assessed the quality of the new version of the MS before looking at the previous reviewer's comments and responses.

My overall view is that this work should definitely be published and I think Roy Soc Open Sci is a good fit for this MS. The amount of work gone into the study is impressive and the presentation of the pilot studies to support the main experiments is great to see.

I do, however, think that whilst the overall results presented are of interest, there are a number of potential caveats that need to be better considered/explained in the MS so that a lay reader can understand the possibilities of why some studies may have found preferences whilst others (such as yours) may not.

Major Comments

Experiment 1

I think the key methodological differences between the Kessler and Arce papers needs to be made clearer. This is important as your 1st experiment is more akin to the Kessler paper in which solitary individuals are used and a preference after 24 hours is studied. Arce actually found avoidance during the first 24 hours and it was only after a number of days and repeated visits (i.e. in the tens of visits per bee) that colony level preferences were starting to manifest.

In my view there are issues with the design of placing a worker in solitary confinement for 24 hours to choose between two solutions to feed upon (as Kessler and your study conducts). Firstly, solitary workers do not act normally and have little motivation to forage in a normal manner. Secondly, it is not clear to me how much a worker will repeatedly feed during 24 hours, and from what I can tell (sorry if I have misunderstood) you do not provide any behavioural data to show frequency of feeds from the tubes. Therefore, what does your consumption data actually tell you about how many times a worker needs to sample a tube to reinforce a preference or aversion? This data is needed to elucidate whether a worker in fact finds a tube, feeds on it and is then satiated for a long period of the experiment. If feed frequency is low then are you actually testing a preference appropriately? For instance, they may not have had the chance to sample each tube enough times for any preference to be reinforced, and therefore you may expect a 50:50 outcome, which seems to be what you found. I think you therefore need to analyse some of your data to convince the reader this is not the case. Glancing through your raw data you could perhaps use consumption rate differentials per time point to give you a value on this or perhaps look at last feed or at least highlight this potential caveat in the discussion. If you do not agree with me, then I think your design needs to be better justified somewhere in the methods. Third, following on from this, does having a higher % sugar concentration increase the risk of what I have highlighted above i.e. on drinking a high calorie solution (e.g. 30%) may mean that the worker is less likely to drink again than if presented with low calorie (e.g. 5%) because of the energetic benefit. Was this found? Fourth, if your bees are different sizes then this may also determine the frequency of feeds and hence bring variation into your data. When considering points 2-4 are you able to control for these possible confounding effects?

Experiment 2

Arce findings showed that whilst the preference was significant, the majority of visits to neonic feeders was not comparatively huge but perhaps more subtle. My concern with your 2nd experiment is that there are at least two stimuli that are dramatically different between your feeders associated with treated or untreated solutions. Is it therefore not possible that these conflicting cues could override any avoidance/preference? Also, is six hours enough time to develop this preference and can you add frequency of visits and time per hour for each of the 1st, 2nd and 3rd presentations? You use your Exp S4 to justify, but the difference between 10 and 50% is quite large and a better justification for being able to associate cues with a post-ingest effect using this setup may need to be more clearly explained.

Minor

Line 34 – insert over a 24-hour period.

Line 51 – ‘after exposure bees forage less frequently’ – I am not sure precisely what you mean by this statement.

Line 264-265 – does this suggest you have stressed the bee?

Lines 347-350 – would this not be better placed in the intro?

Line 353 – you use the term doses, but I think it is more accurate you use concentration because you did not consider this as a covariate in your analyses. i.e. consumption per unit gram of mass would better give dosage.

Line 358 – be clear that Arce had a very different experimental setup.

Reviewer: 2

Comments to the Author(s)

I feel the authors have done a good job of addressing the initial reviewers' concerns with the statistics. It all made sense to me. I see no problems. My assessment is that this paper makes a very important contribution. They have not been able to find a preference for neonicotinoids in *B. impatiens*. That places an informative caveat around the generality of the reports for *B. terrestris*. I feel the discussion in this paper is careful and balanced and demonstrates thorough and sound scholarship. I hope this work will inspire further work into this important issue.

My only concern is very minor. It would seem in revision that some of the reference formatting has gone a bit haywire. The reference format is as a numbered list, but at various points we have references in the format Arce et al (2018). I picked these up at lines 82, 85, 358 and 408, but there may be others. I recommend a careful check of the references before final submission Overall I found this a solid and constructive study.

Author's Response to Decision Letter for (RSOS-191883.R0)

See Appendix A.

RSOS-191883.R1 (Revision)

Review form: Reviewer 1

Is the manuscript scientifically sound in its present form?

Yes

Are the interpretations and conclusions justified by the results?

Yes

Is the language acceptable?

Yes

Do you have any ethical concerns with this paper?

No

Have you any concerns about statistical analyses in this paper?

No

Recommendation?

Accept with minor revision (please list in comments)

Comments to the Author(s)

I appreciate the responses made by the authors to my previous comments. I think the manuscript is acceptable for publication.

But I think integrating some of your answers to my comments into the discussion would improve the paper and increase the transparency of the work done.

I would also advise to try and address my questions concerning experiment 2 and place this in the discussion.

Decision letter (RSOS-191883.R1)

06-Apr-2020

Dear Professor Muth,

On behalf of the Editors, I am pleased to inform you that your Manuscript RSOS-191883.R1 entitled "No evidence for neonicotinoid preferences in the bumblebee *Bombus impatiens*" has been accepted for publication in Royal Society Open Science subject to minor revision in accordance with the referee suggestions. Please find the referees' comments at the end of this email.

The reviewers and Subject Editor have recommended publication, but also suggest some minor revisions to your manuscript. Therefore, I invite you to respond to the comments and revise your manuscript.

- Ethics statement

- Data accessibility

<http://datadryad.org/submit?journalID=RSOS&manu=RSOS-191883.R1>

- Competing interests

- Authors' contributions

All submissions, other than those with a single author, must include an Authors' Contributions section which individually lists the specific contribution of each author. The list of Authors

should meet all of the following criteria; 1) substantial contributions to conception and design, or acquisition of data, or analysis and interpretation of data; 2) drafting the article or revising it critically for important intellectual content; and 3) final approval of the version to be published.

- Acknowledgements

- Funding statement

Because the schedule for publication is very tight, it is a condition of publication that you submit the revised version of your manuscript before 15-Apr-2020. Please note that the revision deadline will expire at 00.00am on this date. If you do not think you will be able to meet this date please let me know immediately.

- 1) A text file of the manuscript (tex, txt, rtf, docx or doc), references, tables (including captions) and figure captions. Do not upload a PDF as your "Main Document".
- 2) A separate electronic file of each figure (EPS or print-quality PDF preferred (either format should be produced directly from original creation package), or original software format)
- 3) Included a 100 word media summary of your paper when requested at submission. Please ensure you have entered correct contact details (email, institution and telephone) in your user account
- 4) Included the raw data to support the claims made in your paper. You can either include your data as electronic supplementary material or upload to a repository and include the relevant doi within your manuscript

5) All supplementary materials accompanying an accepted article will be treated as in their final form. Note that the Royal Society will neither edit nor typeset supplementary material and it will be hosted as provided. Please ensure that the supplementary material includes the paper details where possible (authors, article title, journal name).

Kind regards,

on behalf of Dr Jake Socha (Associate Editor) and Kevin Padian (Subject Editor)
openscience@royalsociety.org

Associate Editor Comments to Author (Dr Jake Socha):

Thank you for the considerable changes you have made to improve the manuscript, which is now acceptable for publication. However, I would like you to strongly consider the final comments from Reviewer 1, as I agree that incorporating some of your responses into the Discussion would increase the quality and transparency of the paper. As you see fit, please revise the manuscript accordingly.

Reviewer comments to Author:

Reviewer: 1
Comments to the Author(s)

I appreciate the responses made by the authors to my previous comments. I think the manuscript is acceptable for publication.

But I think integrating some of your answers to my comments into the discussion would improve the paper and increase the transparency of the work done.

I would also advise to try and address my questions concerning experiment 2 and place this in the discussion.

Author's Response to Decision Letter for (RSOS-191883.R1)

See Appendix B.

Decision letter (RSOS-191883.R2)

14-Apr-2020

Dear Professor Muth,

It is a pleasure to accept your manuscript entitled "No evidence for neonicotinoid preferences in the bumblebee *Bombus impatiens*" in its current form for publication in Royal Society Open Science. The comments of the reviewer(s) who reviewed your manuscript are included at the foot of this letter.

on behalf of Dr Jake Socha (Associate Editor) and Kevin Padian (Subject Editor)
openscience@royalsociety.org

Associate Editor Comments to Author (Dr Jake Socha):

Thank you for addressing the reviewer's comments further, and congratulations on this publication and interesting contribution on neonicotinoid-related mysteries.

Appendix A

Dear Andrew Dunn,

On behalf of myself and my co-authors, we thank you, Dr Socha and Dr Padian for considering our manuscript “*No evidence for neonicotinoid preferences in the bumblebee *Bombus impatiens*” for publication in *Royal Society Open Science*. We are grateful to the two reviewers for their helpful comments on our manuscript. We are glad that Reviewer 2 felt that we had answered the original reviewers’ queries, and below we have addressed the questions and suggestions of Reviewer 1. To make it easier to locate the edits we have made, the new additions are also highlighted in the main manuscript text.*

Kind regards,

Felicity Muth

Reviewer: 1

Comments to the Author(s)

I would first like to disclose that I was not an original reviewer of the previous MS submission. In an attempt to be as unbiased as possible, I assessed the quality of the new version of the MS before looking at the previous reviewer’s comments and responses.

My overall view is that this work should definitely be published and I think Roy Soc Open Sci is a good fit for this MS. The amount of work gone into the study is impressive and the presentation of the pilot studies to support the main experiments is great to see.

I do, however, think that whilst the overall results presented are of interest, there are a number of potential caveats that need to be better considered/explained in the MS so that a lay reader can understand the possibilities of why some studies may have found preferences whilst others (such as yours) may not.

Major Comments

Experiment 1

I think the key methodological differences between the Kessler and Arce papers needs to be made clearer. This is important as your 1st experiment is more akin to the Kessler paper in which solitary individuals are used and a preference after 24 hours is studied. Arce actually found avoidance during the first 24 hours and it was only after a number of days and repeated visits (i.e. in the tens of visits per bee) that colony level preferences were starting to manifest.

We thank the reviewer for pointing out this important point, and have now clarified this in the text (lines 66-75).

In my view there are issues with the design of placing a worker in solitary confinement for 24 hours to choose between two solutions to feed upon (as Kessler and your study conducts). Firstly, solitary workers do not act normally and have little motivation to forage in a normal manner.

We agree that when an individual forager is away from its colony that it may not forage in the same way as it would when foraging for a colony. Using the same protocol we have found that bees do show clear preferences where we would expect them (e.g. between different concentrations of sucrose solution, unpublished data). However, we agree that there could still be differences between how an individual forages for itself vs. for a colony. We have added a sentence to this effect in the discussion (lines 447-451).

Secondly, it is not clear to me how much a worker will repeatedly feed during 24 hours, and from what I can tell (sorry if I have misunderstood) you do not provide any behavioural data to show frequency of feeds from the tubes. Therefore, what does your consumption data actually tell you about how many times a worker needs to sample a tube to reinforce a preference or aversion? This data is needed to elucidate whether a worker in fact finds a tube, feeds on it and is then satiated for a long period of the experiment. If feed frequency is low then are you actually testing a preference appropriately? For instance, they may not have had the chance to sample each tube enough times for any preference to be reinforced, and therefore you may expect a 50:50 outcome, which seems to be what you found. I think you therefore need to analyse some of your data to convince the reader this is not the case. Glancing through your raw data you could perhaps use consumption rate differentials per time point to give you a value on this or perhaps look at last feed or at least highlight this potential caveat in the discussion. If you do not agree with me, then I think your design needs to be better justified somewhere in the methods.

Unfortunately we do not have behavioural data for these experiments on whether bees repeatedly sample solutions over the 24 hour period. However, by looking at the data over time, it is clear that bees do not simply go to one feeding tube and only feed from that tube. For example, at 90 minutes (the first time point we analysed), of the 338 bees measured at this time, in only 18 cases did bees consume one solution without having consumed any of the other (e.g. consumption value minus evaporation is less than 0).

We used this method of measuring preference between two solutions because it has shown to be able to detect preferences in other cases (e.g. the similar methods used in Kessler et al. 2015), and we have confirmed in our own lab that preferences can be seen between different concentrations of sucrose (unpublished data). In our current experiment we also found preferences against the neonicotinoid when it was in water. We acknowledge that bees may not sample between tubes as they would when free-foraging and this is discussed on lines 447-451.

Sampling behaviour in preference tube vs. free-flying assays is actually the focus of a forthcoming paper (Richman, Muth and Leonard, in prep.).

Third, following on from this, does having a higher % sugar concentration increase the risk of what I have highlighted above i.e. on drinking a high calorie solution (e.g. 30%) may mean that the worker is less likely to drink again than if presented with low calorie (e.g. 5%) because of the energetic benefit. Was this found?

We did find differences in the total amount consumed across solutions that varied in their sucrose concentration (e.g. see Fig. S2 and line 239 onwards). It is true that if bees needed a higher number of sampling opportunities to show preferences between treatments then this could potentially explain differences between sucrose concentrations, if we had found them. However, because we did not find preferences at any sucrose concentrations, this is not an issue here.

Fourth, if your bees are different sizes then this may also determine the frequency of feeds and hence bring variation into your data. When considering points 2-4 are you able to control for these possible confounding effects?

We did not measure bees in all experiments, but since we represented treatments across testing days and colonies we do not have reason to believe that there would be differences in bee size across treatments. For experiments 1 and S3 we did measure bee body size (head width and inter-tegular width) after the experiment and did not find differences between treatments. This was not included in the final manuscript (due to the large number of experiments/ analyses already!) but we would be happy to include this if the editor wishes.

Experiment 2

Arce findings showed that whilst the preference was significant, the majority of visits to neonic feeders was not comparatively huge but perhaps more subtle. My concern with your 2nd experiment is that there are at least two stimuli that are dramatically different between your feeders associated with treated or untreated solutions. Is it therefore not possible that these conflicting cues could override any avoidance/preference? Also, is six hours enough time to develop this preference and can you add frequency of visits and time per hour for each of the 1st, 2nd and 3rd presentations? You use your Exp S4 to justify, but the difference between 10 and 50% is quite large and a better justification for being able to associate cues with a post-ingest effect using this setup may need to be more clearly explained.

Minor

Line 34 – insert over a 24-hour period.

We have made this change (line 34).

Line 51 – ‘after exposure bees forage less frequently’ – I am not sure precisely what you mean by this statement.

We have now clarified this sentence (line 52).

Line 264-265 – does this suggest you have stressed the bee?

Referring to the sentence “Finally, to address the possibility that preferences at 90 minutes (the first time point at which bees had consumed enough to be statistically assessed using a parametric model) might be masked”. It is not clear why there was variability between bees (within a single treatment group) in whether and how much they consumed up to 90 minutes. Since bees had 2 hours to habituate, we do not think it is likely that it was due to stress, but think that it is more likely that some bees did not discover the feeding tubes until later than other bees.

Lines 347-350 – would this not be better placed in the intro?

We have now added a sentence to this effect in the introduction (line 47).

Line 353 – you use the term doses, but I think it is more accurate you use concentration because you did not consider this as a covariate in your analyses. i.e. consumption per unit gram of mass would better give dosage.

We thank the reviewer for pointing this out and agree that ‘dose’ is confusing here. We have changed this accordingly (line 359-360).

Line 358 – be clear that Arce had a very different experimental setup.

We have now removed the reference to the Arce paper here, since it is not relevant to the discussion that follows, and explained the experimental setup more thoroughly in the introduction (lines 66-75).

Reviewer: 2

Comments to the Author(s)

I feel the authors have done a good job of addressing the initial reviewers' concerns with the statistics. It all made sense to me. I see no problems. My assessment is that this paper makes a very important contribution. They have not been able to find a preference for neonicotinoids in *B. impatiens*. That places an informative caveat around the generality of the reports for *B. terrestris*. I feel the discussion in this paper is careful and balanced and demonstrates thorough and sound scholarship. I hope this work will inspire further

work into this important issue.

My only concern is very minor. It would seem in revision that some of the reference formatting has gone a bit haywire. The reference format is as a numbered list, but at various points we have references in the format Arce et al (2018). I picked these up at lines 82, 85, 358 and 408, but there may be others. I recommend a careful check of the references before final submission Overall I found this a solid and constructive study.

We thank the reviewer for their positive comments. As for the reference formatting, in the examples they give we intended for the reference to be written out (rather than listed as a number), to make it easier for the reader to know which paper we were referring to. However, we would defer to the editor if this goes against journal style formatting.

Appendix B

Dear Andrew Dunn,

On behalf of myself and my co-authors, we thank you, Dr Socha and Dr Padian for considering our manuscript “*No evidence for neonicotinoid preferences in the bumblebee *Bombus impatiens*” for publication in *Royal Society Open Science*. We have now addressed reviewer 1’s comments in the discussion of our manuscript, specifically:*

- On lines 298-302 we now address why we chose multiple cues for the stimuli used in experiment 2, and address the concern that a stimulus preference might override a solution preference.
- On line 320 we clarify that we only filmed the final test trial, but not the stimulus presentation trials. As such, we unfortunately cannot present any data relating to bee behavior during the stimulus presentation trials.
- On lines 331-335 we address the concern that 6 hours might not be sufficient time to detect a post-ingestive preference in our protocol and justify the use of this protocol.

We hope that our manuscript is now suitable for publication.

Kind regards,

Felicity Muth